# Case Report of Serum Sickness-like Reaction following the First Dose of the Chimpanzee Adenovirus-Vectored AstraZeneca COVID-19 Vaccine, ChAdOx1

**DOI:** 10.3390/vaccines11020467

**Published:** 2023-02-17

**Authors:** Areej Awad Alzaidi, Arwa Awad Alzaidi, Modhi Thaiban AlOtaibi, Reem M. Alsheikh

**Affiliations:** 1Family Medicine, Family Medicine Department, Ministry of the National Guard—Health Affairs, College of Medicine, King Saud Bin Abdulaziz University for Health Sciences, King Abdullah International Medical Research Center, Jeddah 21423, Saudi Arabia; 2Medical Laboratory Specialists, Maternity and Children Hospital, Ministry of Health, Makkah 24246, Saudi Arabia; 3Medical Laboratory Specialists, Makkah 24246, Saudi Arabia

**Keywords:** COVID-19, serum sickness, immune-complex-mediated hypersensitivity reaction

## Abstract

Serum sickness-like reaction from serum sickness is critical. Serum sickness-like reaction has comparable symptoms to serum sickness, but their underlying pathophysiology is distinct. This delayed hypersensitivity response was first characterized as a drug-induced reaction and is uncommon in adults; it is more common in children. COVID-19 vaccinations are now being routinely given in the COVID-19 period, and adverse reactions to immunization have been recorded. We present a case of COVID-19 vaccination-induced serum sickness-like reaction which developed after receiving the first dose of AstraZeneca COVID-19 vaccine.

## 1. Background

Serum sickness is an immune-mediated hypersensitivity event. Fever, rash, polyarthritis, or polyarthralgia are common symptoms. It was originally recognized as a distinct entity in patients who had received heterologous antisera in the early 1900s, which had previously been employed to treat infectious disorders. Symptoms normally emerge one to two weeks following exposure to the causative chemical and subside after a few weeks [1]. It is a self-limiting condition with a good prognosis. Serum sickness normally resolves on its own and has no long-term consequences. As yet, no large-scale research has demonstrated any long-term repercussions of serum sickness. However, persistent exposure to a causal substance, leading to several bouts of serum sickness, induced renal failure and mortality in animal models [2]. Vaccination is regarded as an effective means of minimizing viral transmission and illness severity based on current facts and expertise. As a consequence, various vaccinations with “emergency use authorization” have been utilized globally. The chimp adenovirus-vectored vaccine (ChAdOx1 nCoV-19; AZD1222, AstraZeneca) has been utilized in several countries [3]. Despite the virus’s worldwide dissemination, it is thought that a considerable section of the population in many nations has avoided infection and is still vulnerable to SARS-CoV-2 [4]. Vaccines can increase population immunity, prevent major illnesses, and help relieve the present health crisis. The ChAdOx1 nCoV-19 vaccine (AZD1222) was developed at Oxford University in response to increased global efforts to develop and test vaccines against SARS-CoV-2. It consists of the SARS-CoV-2 structural surface glycoprotein antigen (spike protein; nCoV-19) gene in a replication-deficient chimp adenoviral vector ChAdOx1. Anaphylaxis to immunizations is rare, and it usually affects those who have previously been allergic to the vaccine’s components [5]. Allergies following vaccination may be caused by antigens, adjuvants, preservatives, stabilizers, emulsifiers, residual antibiotics, leached packaging components, cell culture materials, and inactivating substances. We describe a case of Serum sickness in a female after the first dose of the Oxford-AstraZeneca COVID-19 vaccine.

## 2. Case Presentation

A 30-year-old previously healthy female nonsmoker presented to the emergency department of the National Guard Hospital, Western Region, Saudi Arabia, with complaints of fever (body temperature, 39 °C), accompanied by emesis several times, generalized malaise, severe myalgia, and polyarticular arthralgia, which prevented her from performing her daily routine activities.

Three days previously, the patient had received the first dose of an inactivated COVID-19 vaccine (the ChAdOx1 nCoV-19 vaccine (AZD1222) developed at Oxford University) without experiencing any immediate adverse reaction. She experienced migrating shoulder, knee, wrist, and interphalangeal joint symptoms, which were accompanied by mild weakening but no redness, edema, or morning stiffness. She had a non-pruritic, petechial rash on her face, torso, and upper and lower limbs. On day 10 she developed a broad non-pruritic petechial maculopapular erythematous rash that occurred arbitrarily and without a pattern over her face, trunk, and the upper and lower extremities. There was no history of gastrointestinal complaints, palpitations, night sweats, or weight loss other than vomiting. The patient did not have a history of dyspnea, sinusitis, epistaxis, hemoptysis, ophthalmic symptoms, oral or genital ulcers, haematuria, or a change in the quantity or color of urine at this time. The patient had never been infected with COVID-19, and never received any vaccine other than ChAdOx1 before. She denied having ever had a history of a respiratory illness, as well as any family history, drug use, allergies, or blood transfusions.

### 2.1. Examination

The patient presented as sick, pale, and fatigued but alert throughout the examination. Afebrile temperature was 37.4 degrees Celsius. Oxygen saturation SpO2 was at 100% at rest. Her resting blood pressure was 110/63 mmHg, her breathing rate was 18, and her heart rate was 109 beats per minute. There was a non-blanching petechial rash over the face, trunk, and lower and upper limb extremities (Figure 1A,B). Neurological and musculoskeletal examinations were unremarkable. Chest examination revealed equal bilateral air entry, no added sound, and negative for cervical lymphadenopathy. The rest of the physical evaluation was otherwise unremarkable.

### 2.2. Investigations

Hemoglobin levels in the blood were measured and found to be (14.3 g/dL; normal range, 13.5~17.5 g/dL), which is within the normal range. Decreased white blood cells of (2.1 g/dL; normal range, 4~11 g/dL), with low neutrophil (0.71 g/dL; normal range, 2.00~7.50 g/dL), and lymphocyte (1.05 g/dL; normal range, 1.5~4.0 g/dL) count. There were no abnormal findings in the coagulation profile or blood smear.

Several inflammatory indicators were beyond their normal ranges, including the erythrocyte sedimentation rate (30 mm/h; normal range, 4~20 mm/h), C-reactive protein (22.6 mg/L; normal range, 0~5 mg/L), and ferritin (496.39 ng/mL; normal range, 15~120 ng/mL). Antinuclear antibody (ANA) was found to be positive in an immunofluorescence test, even though the reference range is negative. Anti-Smith (SM) antibody, anti-Sjogren’s Syndrome A and anti-Sjogren’s Syndrome B, anti-Jo-1 (histidyl-tRNA synthetase), anti-Ribonucleoprotein, DNA double-stranded antibody, antineutrophil cytoplasmic autoantibody, perinuclear (p)-ANCA, antineutrophil cytoplasmic autoantibody, cytoplasmic (c)-ANCA, and rheumatoid factor were all negative. Complement levels, including C3 and C4, were normal.

Urinalysis, serum creatinine, and blood urea nitrogen results were normal. Transaminases showed elevated Alanine transaminase (ALT) (113 U/L; normal range, 7 to 55 U/L). and Aspartate transaminase (AST) (193 U/L; normal range, 8 to 48 U/L).

Serology for hepatitis B and C viruses was not detected. Positive IgG tests for viruses such as hepatitis A, cytomegalovirus, Epstein–Barr virus, and Brucella indicated a previous infection. Negative results were found in a QuantiFERON-TB assay, a Syphilis total antibody serology, and a SARS-CoV-2 infection real-time polymerase chain reaction. An X-ray of the chest revealed no abnormalities.

### 2.3. Treatment

For further evaluation, the patient was admitted in the emergency department for observation and monitoring. Corticosteroid injection (methylprednisolone succinate formulation 60 mg/day) was given in addition to once-daily antihistamines (Diphenhydramine 50 mg PO) and a moderate-potency topical steroid therapy (Betamethasone valerate 0.1% CREAM). The patient’s symptoms improved progressively in less than two weeks after being discharged on day two, including: the fever subsided, skin rash and joint pains disappeared.

### 2.4. Follow-Up and Outcome

The patient had been clinically stable and was periodically monitored. When inflammatory indicators returned to normal a month later, the physician suggested avoiding receiving a second dose of the same vaccination.

## 3. Discussion

Serum sickness syndrome is a hypersensitivity response caused by an immune complex that arises after immunization and protein-based drugs [6]. Serum sickness-like reaction, on the other hand, despite its clinical similarities to serum sickness syndrome, has an unknown pathophysiology, while recent research shows that it is not driven by aberrant immune-complex formation [7]. Differentiating Serum sickness-like reaction from serum sickness is critical [8,9]. Serum sickness-like reaction has comparable symptoms to serum sickness, but their underlying pathophysiology is distinct [10]. This delayed hypersensitivity response was first characterized as a drug-induced reaction and is uncommon in adults; it is more common in children, with a frequency of around 7% [7]. A comparison of the main characteristics of serum sickness versus serum-sickness like reaction is depicted in Table 1.

Without cutaneous or systemic vasculitis, the triad of rash, fever, and arthralgia constitutes a Serum sickness-like reaction. Urticarial rashes with other systemic symptoms raise suspicion for systemic sclerosis leprosy. Hypocomplementemia, nephropathy, and vasculitis are all symptoms of serum sickness, a type 3 hypersensitivity reaction, but a Serum sickness-like reaction does not [11]. Serum sickness-like reaction rashes are nonspecific and may include urticaria, morbilliform eruption, and polycyclic plaques [7]. Histopathology of the rashes often indicates neutrophilic urticaria without vasculitis [12,13].

Antibiotics (particularly beta-lactams), anticancer, anticonvulsant, antidepressant, antidysrhythmic, antihypertensive, and NSAIDs are only some of the non-protein medicines that may cause Serum sickness-like reaction, which is clinically comparable to the classical version. Serum sickness-like reaction is caused by bacterial and viral infections [14,15].

Serum sickness syndrome and urticarial vasculitis might be difficult to distinguish from serum sickness-like reactions. Though erythrocyte extravasation raises the possibility of urticarial vasculitis, limited perivascular leukocytoclastic has been documented in a prior case of serum sickness-like reaction, and therefore does not necessarily suggest the existence of vasculitis [16]. Joint involvement is also more common in hypocomplementemic urticarial vasculitis than in normocomplementemic urticarial vasculitis [17,18,19,20]. Furthermore, although not all viral infections are studied, these diagnoses are uncommon since they are frequently accompanied by additional viral infection-specific symptoms (e.g., transaminitis for viral hepatitis and pharyngitis for Epstein–Barr virus infection).

Antibiotics, psychiatric medicines (mainly bupropion), biologics, and vaccinations have all been linked to serum sickness-like reactions in adults. Serum sickness-like reactions have been associated with influenza, hepatitis B, and rabies vaccinations [21,22,23]. A 43-year-old female patient with a serum sickness-like reaction after receiving the first dose of an inactivated COVID-19 vaccine (CoronaVac; Sinovac, Beijing) was reported by Chaijaras et al. [20]. In addition to severe myalgia and arthralgia, the rashes were accompanied by fever, generalized malaise, and cervical lymphadenopathy [24]. Our case adds to evidence of inactivated COVID-19 immunization to the list of disease triggers; serum sickness-like reactions linked to this vaccine may be characterized as hypersensitivity responses [25]. A case of polymyositis secondary to an autoimmune reaction following vaccination was reported by Capassoni et al. [26] in which the patient developed a skin rash (characterized by plaques of vesicular, vesicular-papular, partially confluent lesions with peripheral halo and necrotic center on thighs, feet, and face) and myositis in the tibialis anterior muscle after the AstraZeneca. Recognizing this disease is critical because it prevents patients from getting more doses of vaccination unless an essential requirement outweighs the danger of re-developing this illness. 

Serum sickness-like reactions usually resolve on their own within a week or two after the triggering factors have been eliminated [6], however, this time frame might be extended to three weeks in rare cases. Patients with severe illness, on the other hand, may need non-steroidal anti-inflammatory drugs or corticosteroid therapy [27].

Pain and soreness at the injection site, fever, chills, malaise, weariness, myalgia, arthralgia, nausea, and headache were the most frequently reported adverse events throughout trials of this vaccine in the United Kingdom, Brazil, and South Africa. Anaphylaxis, loss of appetite, lymphadenopathy, dizziness, and stomach discomfort were observed in very small numbers [28]. A 62-year-old lady with a history of paraesthesia and numbness in both lower limbs for 3 days arrived at the urgent medical unit with worsening weakness. She had her first injection of the Oxford/AstraZeneca COVID-19 vaccination 11 days before presenting with symptoms [29]. The first reported instance of Guillain–Barré syndrome (GBS) after administration of the Pfizer-BioNTech COVID-19 vaccination was reported by Waheed et al. [30]. One incidence of GBS has been documented after the introduction of the vaccine developed by Oxford University and AstraZeneca [31]. Siddig A reported three instances of thrombosis with thrombocytopenia after the introduction of the adenovirus vector vaccine ChAdOx1 (Oxford-AstraZeneca), all of which manifested between 8 and 14 days after the first vaccination. Ischemic stroke, cerebral venous thrombosis, and particularly sagittal sinus thrombosis were indicated in this research as the most common kinds of stroke that are likely to emerge after immunization [32]. 

A case report by Muath et al. describes a 43-year-old who, three days after getting the first dose of the AstraZeneca vaccine against COVID-19, exhibited right-sided abrupt onset, upper and lower limb paralysis with the Ischemic stroke diagnosis [33].

Folegatti PM et al. reported a satisfactory safety profile and enhanced antibody responses of ChAdOx1 nCoV-19 in a phase 1/2, single-blind, randomized controlled study in five UK locations [34]. 

ChAdOx1 nCoV-19 was reported by Frater J et al. to be safe and immunogenic in a 2/3 trial on HIV patients, recommending immunization for individuals receiving antiretroviral medication with good control. People with HIV and HIV-negative individuals were recruited in the phase 1B/2A study and randomly assigned at seven South African sites [35]. ChAdOx1 nCoV-19 was well tolerated in this study, indicating favorable safety and immunogenicity in patients with HIV, including increased immunogenicity in SARS-CoV-2 baseline-seropositive subjects, according to Madhi SA [36]. The trials conducted in the UK, Brazil, and South Africa were reported by Voysey M et al. In an interim review of current clinical studies, DChAdOx1 nCoV-19 was reported to be effective against symptomatic COVID-19 and to have a tolerable safety profile.

One limitation of the research is that we were unable to do a skin biopsy of the rash because the patient refused at that time [37]. 

## 4. Learning Points

Serum sickness-like reaction is a rarely reported side effect and risk following AstraZeneca COVID-19 vaccine, but benefits of currently approved vaccines significantly outweigh their risks.We describe a first case in Saudi Arabia of serum sickness-like reaction following vaccination against COVID-19, in the absence of any further triggers.Further association studies are required to adequately determine the potentially link between COVID-19 vaccination, serum sickness-like reaction.Serum sickness-like reactions is a self-limiting illness usually resolve on their own within a week or two after the triggering factors have been eliminated, however, this time frame might be extended to three weeks in rare cases.

## 5. Conclusions

The present case highlights a serum sickness-like response linked with the COVID-19 vaccine; doctors should be aware of this reaction type after immunization, despite its rarity. However, there should not be any doubt about the COVID-19 vaccine because of its side effects. 

## Figures and Tables

**Figure 1 vaccines-11-00467-f001:**
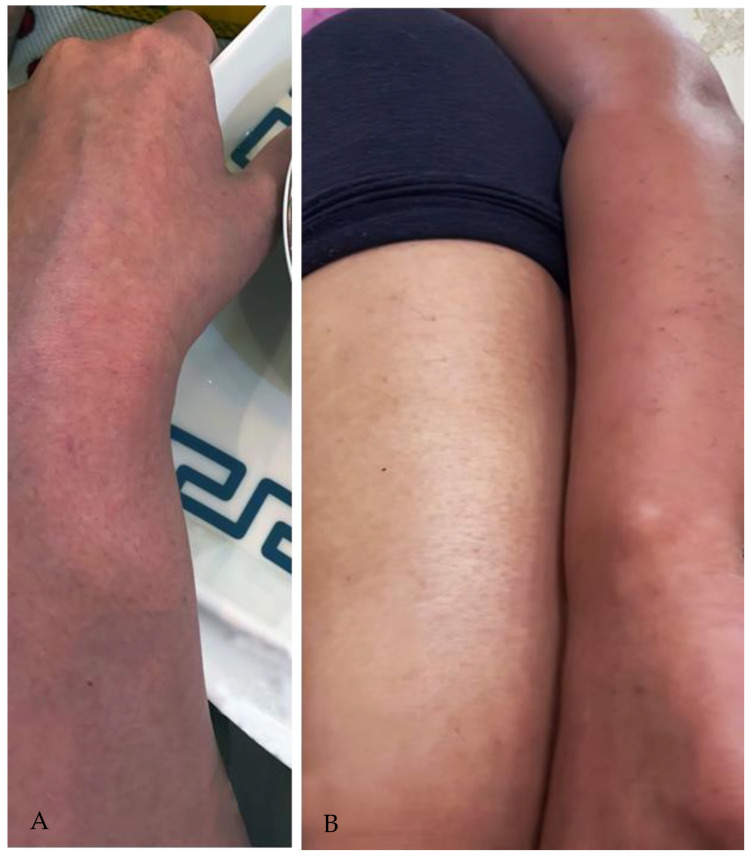
(**A**,**B**) a non-blanching petechial rash over the Lower and upper limb extremities.

**Table 1 vaccines-11-00467-t001:** Comparison of the clinical and pathophysiological characteristics of serum sickness-like reaction versus serum sickness [10,11,12,13,14,15].

**Condition**	**Serum Sickness-like Reaction**	**Serum Sickness**
**Causal agents**	- Chemical drugs, such as anticonvulsants (phenytoin), antibiotics (most commonly cefaclor, penicillin, and trimethoprim-sulfamethoxazole) NSAIDs, - Bacterial or viral agents such as streptococcus and hepatitis B virus, - Some vaccines such as rabies vaccines, and less frequently influenza and tetanus	Agents containing heterologous (nonhuman) proteins or antigens, such as vaccines, biological therapies (immune modulators, monoclonal antibodies), equine microbial or venom anti-toxins, insect proteins, etc.
**Pathophysiology**	The pathophysiology is still not elucidated. It is believed that the mechanism is drug-specific and may be due to: (1) a direct toxicity of the drug or its metabolites on immune cells, or (2) drug-induced immune complexes, where the drug molecules (or its metabolites) bind to serum proteins triggering an antibody response with a hypersensitivity reaction. (3) Genetic susceptibility	The pathophysiology involves type III (immune complex-mediated) hypersensitivity mechanism. Exposure to heterologous proteins induces an immune response with the formation antigen-antibody immune complexes. In case of reduced clearance (due to reduced function of macrophage activating system), these immune complexes settle in body tissues and joint fluids, which leads to the activation of the complement. The complement activation stimulates a histamine-mediated local inflammatory response causing the symptoms. Other mechanisms suggest the presence of complement-independent pathways for serum sickness involving the reaction of the immune complex with Fc-gamma-receptors that present on immune cells, such as neutrophils.
**Clinical presentation**		
*Time from exposure to onset*	5–10 days	6–10 days after first exposure to the antigen, and fewer days in case of repeat exposure.
*Dermatological manifestations*	Rash made of urticaria and pruritis	Urticaria, maculopapules or purpura (vasculitis). Mucus membranes are not involved.
*Rheumatic manifestations*	Arthralgias, and less commonly arthritis.	Polyarthralgia and less commonly polyarthritis, involving more frequently knees, shoulders and metacarpophalangeal articulations.
*Systemic and other disorders*	Fever is frequent but inconsistent and lymphadenopathies are less common.	Fever and malaise are frequent. Less common manifestations are: edema, lymphadenopathies, visual disorders, splenomegaly, nephropathy, neuropathy, etc.
*Prognosis*	Resolution of major symptoms (fever, arthralgias) ~2 days after discontinuation of the causal agent.	Resolution 7–14 days after discontinuation of the causal agent. Multiples episodes due to repeat exposures to the causal agent may lead to renal failure and death.
**Biological findings**	- Leukocytosis - Elevated erythrocyte sedimentation rate - Normal serum complement levels	- Neutropenia - Mild proteinuria in urinalysis (50% of the patients), with mild hematuria - High serum creatinine level - High titers of antibodies - Low levels of circulating C3 and C4 and total hemolytic complement (CH50)

## Data Availability

The data presented in this study are available in the article.

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
