# Peer review of "Case Report of Serum Sickness-like Reaction following the First Dose of the Chimpanzee Adenovirus-Vectored AstraZeneca COVID-19 Vaccine, ChAdOx1"

_vaccines, 2023, doi:10.3390/vaccines11020467_

Round 1

Reviewer 1 Report

SARS-CoV-2 infection is still a global concern and vaccines from different companies are widely used around the world. But side effects have been reported in many cases. In this article, Areej Awad Al Zaidi and colleagues reported a case of serum sickness-like reaction after the first dose of AstraZeneca COVID-19 vaccine. The patient is reported with a non-blanching petechial rash over the face, trunk, Lower and upper limbextremities, increased inflammatory indicators, and elevated transaminases in urinalysis. After 2 days of treatment, the patient was discharged with the fever subsided, skin rash and joint pains disappeared.

Overall, the case report is well-written, and the clinic test approaches and the interpretation of the data are appropriate. But there are several points to be strengthened to make such conclusions. Some major comments/questions are found below:

1: what’s the date of this patient get vaccinated?

2: what’s the percentage of getting AstraZeneca in Saudi Arabia?

3: did the patient get infected by COVID-19 before she got vaccinated?

4: “Serum sickness-like reaction has comparable symptoms to serum sickness, but their underlying pathophysiology is distinct.” What’s the underlying pathophysiology? The authors should explore the mechanism behind it.

5: the authors could summarize a table of differential diagnoses of serum sickness-like reaction with other syndromes.

Author Response

1: what’s the date of this patient get vaccinated?

Added to manuscript 15/3/2021

2: what’s the percentage of getting AstraZeneca in Saudi Arabia?

No data available in our country 

3: did the patient get infected by COVID-19 before she got vaccinated?

No she did not get infected before

Added to manuscript 

4: “Serum sickness-like reaction has comparable symptoms to serum sickness, but their underlying pathophysiology is distinct.” What’s the underlying pathophysiology? The authors should explore the mechanism behind it.

some different pionts discussed in the article 

5: the authors could summarize a table of differential diagnoses of serum sickness-like reaction with other syndromes.

done

Reviewer 2 Report

Thank you for sharing the very well written article on serum sickness-like reaction following vaccination with ChAdOx1.

You report that serum sickness-like reaction (SSR) can occur when certain medication is given. Did you assess whether the 30 year old female case did administer any medication that could coincide with the ChAdOx1 vaccination? You also report that SSR is well known in the context of bacterial and viral infections? You state some viral pathogens assessed, but which bacterial pathogens did you look for in a rather systematic manner? Also, you state some association between SSR and other than COVID-19 vaccines. Did you assess whether the 30 year old female case did receive any other vaccine prior to the ChAdOx1?

Author Response

Did you assess whether the 30 year old female case did receive any other vaccine prior to the ChAdOx1?

She never received any other than ChAdOx1 vaccine before 

Added to manuscript  

Reviewer 3 Report

Estimated Authors,

I've been invited to review this case report from Saudi Arabia, whose content outlines a case of serum sickness in a 30-year-old previously healthy female nonsmoker. The patient has developed a whole spectrum serum sickness (i.e. "rash, fever, and polyarthralgias or polyarthritis, which begin one to two weeks after the first exposure to the responsible agent and resolve within a few weeks of discontinuation"; https://www.uptodate.com/contents/serum-sickness-and-serum-sickness-like-reactions), and even though the delay from the vaccination (i.e. 3 days) is somehow shorter than expected, both the complained signs and symptoms, and the clinical course of the reported case are consistent with the usual definition of Serum Sickness. The temporal sequence between vaccination and clinical presentation is clearly suggestive, but Authors should report in a somehow more cautious way the association between vaccination and this clinical case.

From the point of view of the present reviewer, some further improvements are required and mostly focus on the overall quality of the English text, that is often affected by typos and improper spelling. Some examples:

- "Three days previous; The patient had received the first dose of an inactivated COVID-19 vaccine" (did you mean: "Three days before the onset of the symptoms, the patient had received...") 

- "Chest examination revealed equal bilateral air entry, no added sound. And negative for cervical lymphadenopathy." Maybe: "Chest examination revealed equal bilateral air entry, without added sound. The patients was also negative for cervical lymphadenopathy"

- "Hemoglobin levels in the blood were measured and found to be (14.3 g/dL; normal range, 13.5 ~ 17.5 g/dL)" maybe: "Hemoglobin levels in the blood were measured and found to be 14.3 g/dL (normal range, 13.5 ~ 17.5 g/dL)"

and so on.

Therefore, I'm advocating for a minor revision, mostly focusing on the English editing.

Author Response

I'm advocating for a minor revision, mostly focusing on the English editing

Response:

Manuscript reviewed by native english 

Round 2

Reviewer 1 Report

The authors answered my question well. i don't have more comments.